# Improving Primary Care After Stroke (IPCAS) randomised controlled trial: protocol for a multidimensional process evaluation

Maria Raisa Jessica (Ryc) Aquino [1,2] Ricky Mullis,[1] Elizabeth Kreit,[1] Vicki Johnson,[3] Julie Grant,[1] Lisa Lim,[1] Stephen Sutton,[1] Jonathan Mant[1]

¹Department of Public Health and Primary Care, University of Cambridge, Cambridge, UK
²Population Health Sciences Institute, Newcastle University, Newcastle upon Tyne, UK
³Leicester Diabetes Centre, University Hospital Leicester NHS Trust, Leicester, UK

**Correspondence to**
Dr Maria Raisa Jessica (Ryc) Aquino;
ra532@medschl.cam.ac.uk

## ABSTRACT

**Introduction** Primary care interventions are often multicomponent, with several targets (eg, patients and healthcare professionals). Improving Primary Care After Stroke (IPCAS) is a novel primary care-based model of long-term stroke care involving a review of stroke-related needs, a self-management programme, a direct point of contact in general practice, enhanced communication between care services, and a directory of national and local community services, currently being evaluated in a cluster randomised controlled trial (RCT). Informed by Medical Research Council guidance for complex interventions and the Behaviour Change Consortium fidelity framework, this protocol outlines the process evaluation of IPCAS within this RCT. The process evaluation aimed to explore how the intervention was delivered in context and how participants engaged with the intervention.

**Methods and analysis** Mixed methods will be used: (1) design: intervention content will be compared with 'usual care'; (2) training: intervention training sessions will be audio/video-recorded where feasible; (3) delivery: healthcare professional self-reports, audio recordings of intervention delivery and observations of My Life After Stroke course (10% of reviews and sessions) will be coded separately; semistructured interviews will be conducted with a purposive sample of healthcare professionals; (4) receipt and (5) enactment: where available, structured stroke review records will be analysed quantitatively; semistructured interviews will be conducted with a purposive sample of study participants. Self-reports, observations and audio/video recordings will be coded and scored using specifically developed checklists. Semistructured interviews will be analysed thematically. Data will be analysed iteratively, independent of primary endpoint analysis.

**Ethics and dissemination** Favourable ethical opinion was gained from Yorkshire & The Humber-Bradford Leeds NHS Research Ethics Committee (19 December 2017, 17/YH/0441). Study results will be published in a peer-reviewed journal and presented at relevant conferences.

**Trial registration number** NCT03353519; Pre-results.

### Strengths and limitations of this study

► A strength of this process evaluation protocol includes the use of implementation frameworks, in particular, a multidimensional approach to evaluation and intervention fidelity assessment.
► This study applies quantitative and qualitative methods and an iterative approach to analysis to understand how the intervention is implemented in practice.
► This process evaluation will provide insight about uptake and barriers and facilitators to implementing a new model of care in general practice.
► A limitation of our approach is that interview findings might not be generalisable to other settings.
► We have adapted our interview sampling approach to capture as broad a range of study involvement as possible and aimed to triangulate these data with other data sources, such as observations, questionnaires and routinely collected data.

## INTRODUCTION

Stroke mortality is in decline, but remains a leading cause of disability.[1–3] There are rising numbers of people with stroke living in the community, and with this comes increasing demand on primary care services to address long-term needs, such as cognitive, psychological and social problems.[3 4] Strategies to address these long-term needs are limited, with stroke survivors and their caregivers often reporting a lack of support.[3 5] Stroke prevention is a priority for the National Health Service (NHS) in England,[6] as most recently outlined in the *NHS Long Term Plan*, and internationally.[6–9] Primary care services have tended to focus on secondary prevention of stroke and risk factor management[10 11] rather than addressing the longer-term needs,[5] and no formal primary care-based model of care exists to support community-dwelling stroke survivors.

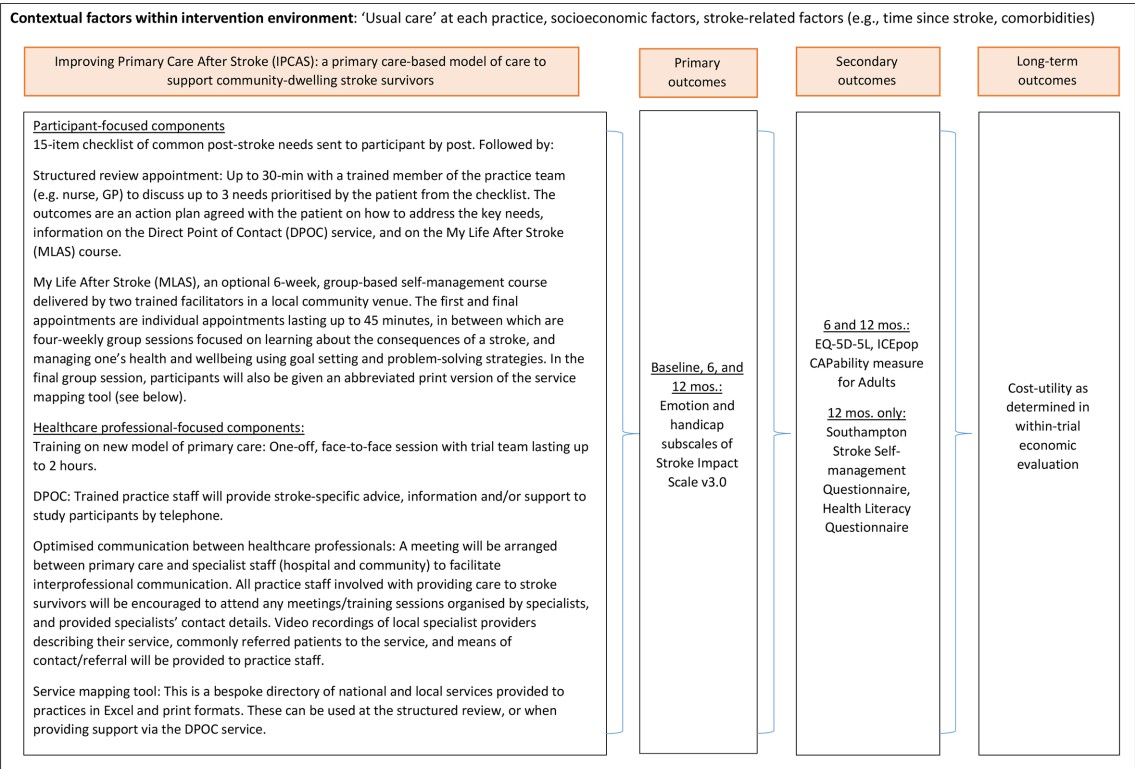

**Figure 1** Interventions and outcomes. GP, general practitioner.

The Improving Primary Care After Stroke (IPCAS) cluster randomised controlled trial (RCT) aims to evaluate the clinical and cost effectiveness of a new model of primary care for stroke survivors living in the community compared with standard care and has been described previously.[12] General practices randomised to the intervention arm will deliver a multicomponent package of care comprising a one-off structured review with a trained primary care professional using a novel needs checklist,[13] a self-management course (My Life After Stroke (MLAS)) conducted by trained facilitators over a 9-week period, which aims to improve confidence, independence and hope in survivors of stroke through problem-solving and experience and knowledge exchange with course participants; enhanced communication between healthcare professionals; and a direct point of contact in primary care accessible to patients for stroke-specific support throughout the study period. General practices randomised to the control arm will provide 'usual care', the details of which will be documented as part of this process evaluation (see the Fidelity of design section). The primary endpoint will be two subscales (emotion and handicap) of the Stroke Impact Scale V.3.0[14] as coprimary outcomes at 12 months after entry into the trial. The intervention and outcomes are summarised in figure 1.

Complex interventions are widely used to address health problems and consist of multiple interacting components,[15 16] often with several targets (eg, patients and healthcare professionals) and implemented in various settings.[17 18] Consequently, variations in implementation could influence intervention effects on outcomes.

Process evaluations are especially important for RCTs, such as IPCAS, which is being conducted in 46 general practices in the East of England and the East Midlands. These practices have contextual differences, and thus, the intervention may undergo adaptations as the study is conducted.[18 19] It also has intervention components that are aimed at stroke survivors, as well as healthcare professionals. This warrants a multifaceted process evaluation to identify successes, failures and/or unintended consequences.[15 20 21] Understanding the delivery of complex interventions is key to determining the feasibility of delivering these in other settings and at scale.[17] As such, the present evaluation was developed alongside IPCAS and will be conducted in parallel with it.

The UK Medical Research Council suggests that process evaluations are conducted to understand how interventions are carried out, how intervention activities produce change and how the context in which interventions are conducted influences delivery.[18 19] In line with this, the US National Institute of Health Behaviour Change Consortium's (BCC) fidelity framework and guidance specify five dimensions of treatment fidelity: (1) design: the degree to which the intervention is distinct (eg, through articulated intervention components and theoretical underpinning) from usual care or other treatment conditions; (2) training: the extent to which the training of intervention providers is adequate through monitoring training processes and activities; (3) delivery: the extent to which intervention components and/or procedures are provided as planned; (4) receipt: the extent to which intervention recipients understand and engage with intervention components; and (5) enactment:

**Table 1** Assessing intervention fidelity in the IPCAS trial

| Fidelity of design | Fidelity of training | Fidelity of delivery | Fidelity of engagement (receipt and enactment) |
|---|---|---|---|
| *Goal* | | | |
| ► To determine the extent to which the intervention reflects its theoretical underpinnings. | ► To determine the extent to which the training curriculum has been provided as planned. | ► To determine the extent to which the intervention was provided as planned. | ► To determine the extent to which stroke survivors understood and applied the skills gained from the intervention. |
| ► To determine the extent to which the intervention is distinct from 'usual care'. | | | |
| *Strategies* | | | |
| ► Coding intervention components (ie, IPCAS training manual and MLAS curriculum) to theoretical underpinnings. | ► Training evaluation forms (MLAS). | ► Audio-recorded observations (IPCAS). | ► Self-report questionnaire (MLAS). |
| ► Comparison of convergence between intervention and control groups (includes recording of participating surgeries' usual care practices). | ► Video-recorded observations (MLAS). <br> ► Audio-recorded observations (IPCAS). <br> ► Postintervention interviews (healthcare professionals, IPCAS). | ► Structured telephone calls to healthcare professionals (IPCAS). <br> ► Direct observations (MLAS). <br> ► Postintervention interviews (IPCAS). | ► Postreview structured telephone calls to participants (IPCAS). <br> ► Postintervention participant interviews. |

IPCAS, Improving Primary Care After Stroke; MLAS, My Life After Stroke.

the extent to which intervention recipients carry out specific/relevant behaviours learnt from the intervention in their daily lives.[22 23]

This paper outlines the aims, design and methods for an embedded process evaluation within an RCT. This process evaluation is a multidimensional assessment of the IPCAS trial, specifically to explore (1) how the intervention was delivered, (2) how participants engaged with the intervention and (3) if/how the context influenced delivery or conduct of the intervention.

## METHODS AND ANALYSIS
### Design
A mixed-methods sequential design will be applied. The process evaluation (encompassing intervention fidelity) will be conducted in parallel to the IPCAS trial. Table 1 outlines how the five BCC fidelity dimensions will be addressed in IPCAS. Each of these dimensions—design, training, delivery, engagement (enactment and receipt)—are discussed in turn, including relevant materials and data collection processes. These are followed by the patient and public involvement (PPI) and analysis plans.

### Fidelity of design
This concerns the completeness of the intervention specification and the degree to which the intervention reflects its theoretical foundations.[20 22] All IPCAS intervention components are defined in the trial protocol,[12] some of which are standardised (eg, structured review checklist for use in a one-off, face-to-face annual stroke review) and some are not, in order to allow participants and practices flexibility to adapt these components to their needs (eg, accessing direct point of contact and MLAS, enhanced communication between healthcare professionals). The intervention development process involved a multidisciplinary team (general practitioners (GPs), patient representatives, primary care experts, applied health researchers, psychologists, stroke physicians, practice nurses and therapists), with pilot work informing the feasibility and acceptability of the intervention.[12 13]

To ensure that the intervention reflects its theoretical underpinnings and is distinct from usual care, the intervention components (eg, structured reviews and MLAS) and parameters (eg, frequency or dose, mode of delivery and duration) will be coded and compared by two independent coders in accordance with established methods.[20 22] As such, all participating general practices will be asked to provide details of usual care processes for stroke care and long-term follow-up, such as the frequency and content of follow-up appointments, healthcare professionals responsible for these, investigations conducted (eg, blood tests, blood pressure and pulse measurements) and other actions taken (eg, referrals or further investigations).

Inter-rater reliability will be assessed using Cohen's kappa (95% CI).[24] Where intervention components of similar frequency and/or mode of delivery are found in both conditions, these will be classed as *fully convergent*; where intervention components are found in both experimental conditions but are of dissimilar frequency and/or mode of delivery, these will be classed as *partially convergent*; and where intervention components are found only in one arm of the study, these will be classed as *unique*.[20] Moreover, adaptations to the intervention will be recorded and monitored by the research team throughout the intervention period through maintaining a written record of these in the fidelity of delivery assessments (detailed in the following subsections).

## Fidelity of training

Participating general practices will be given a 2-hour face-to-face training session where the research team will present the intervention components and facilitate discussions on how to conduct the intervention using fictional vignettes. A sample of the training sessions (20%, up to four sessions) will be audio-recorded, with two raters assessing whether planned components are present using a specifically developed checklist (online supplementary file 1). Randomly-selected training sessions will be audio-recorded (with participant consent) after the fifth training session when trainers have had an opportunity to test their curriculum and materials, and adjustments will be made where necessary. The percentage of present planned components will be calculated, and inter-rater agreement will be assessed using Cohen's kappa (95% CI).[24] MLAS facilitators will be trained via a 3-day workshop, with the research team delivering presentations on stroke and its impact, providing trainees time to practise active listening skills, explaining the philosophy and theories underpinning the self-management course, demonstrating MLAS facilitation and providing an opportunity for trainees to rehearse course delivery. MLAS training will be video-recorded and coded by an independent rater according to a priori criteria (online supplementary file 2) to assess whether planned components are present. In addition, trainees will be encouraged to complete training evaluation forms, to assess facilitator knowledge of the MLAS curriculum and expected facilitator behaviours (online supplementary file 3). The percentage of present planned components will be calculated. To ensure skills are maintained, refresher training sessions will be offered as necessary. MLAS facilitators will receive mentoring from researchers as part of their training package after having had an opportunity to deliver a programme. Qualitative interviews with healthcare providers conducting structured reviews will explore their experiences of intervention training (see the Fidelity of delivery section). Finally, process variables such as the number of healthcare professionals and facilitators trained, and the duration of training sessions will be recorded and summarised.

## Fidelity of delivery

Monitoring intervention delivery is critical to ensuring that the intervention is provided as planned; this will be conducted using quantitative and qualitative methods. The BCC guidance suggests that evaluating intervention sessions by observation (either directly or using audio/video-recording) is the gold standard for monitoring intervention delivery.[22 23] To achieve this, a sample of the structured stroke reviews will be audio-recorded and coded by two independent raters according to a prespecified checklist (online supplementary file 4), which outlines the specific intervention components which healthcare providers are expected to cover. A sample of structured reviews carried out (up to 10% or 46 30-min reviews across all intervention sites) will be audio-recorded at each practice. With participant consent, some reviews will be recorded after the fifth review has been conducted at each practice, which allows for healthcare professionals to rehearse and become familiarised with review components. Fidelity of delivery will be assessed by calculating the percentage of the prespecified structured review content that is present. Inter-rater reliability will be assessed using Cohen's kappa (95% CI).[24] To enable further exploration of delivery, healthcare professionals will be contacted by members of the research team regularly during the intervention period to discuss the structured reviews and to monitor whether the intervention being delivered is as planned, with responses recorded using a prespecified checklist (see online supplementary file 4). These data will be quantified by calculating the percentage of present prespecified structured review content. Additionally, reasons for (eg, lack of time) and the nature of (eg, additional support provided) deviations from the developed intervention will be explored and documented through conducting interviews. A purposively selected sample (approximate n=15, 10 intervention and 5 control) of healthcare professionals will be invited to participate in semistructured interviews to explore their overall experience of delivering the intervention (intervention) or participation in the study (control) in depth. Specifically, interviews will explore healthcare professionals' reflections on their experience of conducting the structured review, the functionality of the direct point of contact and service mapping tool, and their overall experience of the intervention (eg, what worked well vs what did not work well). Interviews will run for approximately 30 min–1 hour. These interviews are planned to take place from the 6-month follow-up period. Recruitment and data collection will cease when adequate data addressing our evaluation aim have been acquired[25 26] (see the Analysis plan section).

MLAS courses will also be assessed for fidelity of delivery. Specific contents of this self-management programme are provided in the IPCAS trial protocol.[12] A sample (ie, two full MLAS courses, approximately 20 hours of observation) of preselected sessions of running MLAS programmes will be directly observed independently by two observers, with disagreements resolved through

consensus discussion. These observers will have been trained in course delivery and observation. A specially developed observation checklist (online supplementary file 5) will be used to code for the presence or absence of intended course contents, materials and facilitator behaviours. Fidelity will be quantified by assessing the proportion of presence of prespecified content (ie, % coded components/planned components). These observations are planned to take place after MLAS facilitators have received mentoring; we also aimed to observe all facilitators running MLAS courses at least once. MLAS facilitator–participant interactions will also be observed using the Diabetes Education and Self-Management for Ongoing and Newly Diagnosed (DESMOND) observation tool (online supplementary file 6), in line with similarly designed courses.[27]

Finally, process variables will be collected to assess delivery and summarised for each participating general practice, such as practice list size, role of healthcare professional conducting reviews, the number of patients attending structured reviews, duration of each structured review conducted, number and content of referrals/signposting to other services and number of MLAS courses conducted. These fidelity of delivery assessments and process measures will offer an insight into differences in intervention delivery between practices.

### Fidelity of receipt and enactment (engagement)

Fidelity of receipt and enactment (engagement), which concerns understanding of intervention content and application of skills gained from the intervention, respectively,[22 28] will be assessed concurrently as both encompass participant engagement[28] both quantitatively and qualitatively. Similar to the process outlined for assessing fidelity of delivery, patients will be contacted by members of the research team to discuss the structured reviews they were invited to. Predetermined questions concerning the structured review will be asked and recorded on a checklist (online supplementary file 7). This will be quantified by calculating the percentage of the present prespecified structured review content. Concerning MLAS, a purposively selected sample of participants (ie, 10% of participants attending MLAS, approximately 20) will be invited to complete a specially developed receipt questionnaire (online supplementary file 8) at the end of each session attended (ie, six per participant in total). If necessary, participants will be offered support in completing this.

Qualitative semistructured interviews with study participants (approximately n=25, 15 intervention and 10 control) will be conducted to explore their understanding and experience of the new model of care (intervention arm), and their care experiences since involvement in the trial (control arm). Participants will be selected using a random purposive method, given the large pool of eligible participants (eg, those who attended structured reviews and MLAS and those who did not), which will allow us to gain representative insights.[29] Interviews will run for approximately 30 min–1 hour. These interviews

are expected to take place from the 6-month follow-up period. Specifically, participants will be asked about any action plans they developed with healthcare professionals during their structured review and the outcomes of these, experiences of accessing the direct point of contact for their needs, as well as participation in MLAS. Interviews with control group participants will focus on any stroke-specific appointments/contacts with the GP or practice staff since involvement in the study and any perceived changes in their care during this period. These different areas of focus will allow for an exploration of differences within and between experimental arms. Recruitment and data collection will cease when adequate data addressing our evaluation aim have been acquired[25 26] (see the Analysis plan section).

In addition, process variables will be collected routinely, such as the number of patients invited and consented, demographic characteristics, time since stroke, number of patients attending structured reviews, number of patients completing the checklist of needs, number of participants enrolled in MLAS (defined as attending the first individual appointment and first group session) and number of participants completing MLAS (defined as attending both individual appointments and at least three of the four group sessions).

### Patient and public involvement

Protocol and materials development involved PPI. A PPI member provided feedback on the process evaluation protocol to ensure procedures (eg, recruitment and data collection) in place are appropriate and relevant. For example, PPI consultation resulted in changes to the format and length of the questionnaires for MLAS, ensuring that the questionnaire is aphasia friendly. Together, we decided to brief facilitators about providing stroke survivors support in completing questionnaires where needed. Our PPI member was also involved in pilot-testing the interview schedule with a member of the research team, which ensured that the interview schedule was suitable and flexible (eg, addition of closed question prompts for people with aphasia).

### Analysis plan

The process evaluation analysis will be undertaken separately to the trial outcome evaluation and will inform the interpretation of these data. Quantitative aspects of the process evaluation (eg, observational data and checklists/questionnaires) will be assessed using descriptive statistics (ie, proportions, frequencies and means) and percentage implementation scores on various BCC fidelity dimensions. This will enable us to examine variation in implementation between GP practices at regular intervals. It is suggested that a score of 80%–100% is representative of 'high' intervention fidelity; 51%–79% is representative of 'moderate' intervention fidelity; and <50% is representative of 'low' intervention fidelity.[30] Qualitative aspects of the process evaluation include audio-recorded interview data and textual data (ie, qualitative data from checklists/

questionnaires). All data will be transcribed verbatim. Interview data will be synthesised using thematic analysis by members of the research team using established methods.[25] This will also involve an iterative comparison of participant and healthcare professional interview data to understand similarities and differences between intervention providers and recipients. Textual data from checklists and questionnaires will be analysed using content analysis. NVivo V.12 will be used to support qualitative data management and analysis.

The quantitative and qualitative analyses will be conducted and integrated at different phases of the evaluation. For example, we will compare findings from different data sources such as participants' self-reported receipt and enactment questionnaires, and healthcare professionals' self-reported delivery questionnaires with the audio-recorded observations of the structured stroke reviews.

### Ethics and dissemination

Favourable ethical opinion for the research was gained on 19 December 2017 from Yorkshire & The Humber-Bradford Leeds NHS Research Ethics Committee. Approval to start was given by the Health Research Authority on 21 December 2017, prior to the recruitment of participants commencing at any NHS site. Process evaluation findings will be published in peer-reviewed journals and presented at relevant academic conferences.

## DISCUSSION

This multifaceted process evaluation aimed to complement the effectiveness testing of a theory-driven complex intervention. It will generate knowledge concerning how and why this novel model of primary care-based stroke care could have an impact, thereby informing ways of scaling up the intervention and implementing the model in other settings as part of usual care. Specifically, the present intervention contains adaptable components as it is implemented pragmatically; therefore, process evaluation findings can offer an insight into how this model of care contributes to the evidence base by illustrating the extent to which a high level of fidelity is useful or necessary.

It has been suggested that there are four possible scenarios as to the contribution of process evaluation findings to investigating how interventions are effective or not[20]: (1) successful intervention/high fidelity, where trial outcomes demonstrate improvements in the intervention group and treatment fidelity is deemed 'high', which increases confidence in associating trial outcomes with the intervention delivered; (2) successful intervention/low fidelity, where trial outcomes demonstrate improvements in the intervention group and treatment fidelity is deemed 'low', which suggests that there may have been confounding factors, or mechanisms of action through which the intervention operated which were not identified and assessed; (3) unsuccessful intervention/

high fidelity: where trial outcomes demonstrate no improvements in the intervention group and treatment fidelity is deemed 'high', which supports the conclusion that the intervention was not effective, or could have been influenced by confounding factors; and (4) unsuccessful intervention/low fidelity: where trial outcomes demonstrate no improvements in the intervention group and treatment fidelity is deemed 'low', which suggests that the intervention was not tested as it was not carried out as planned, or that poor intervention fidelity could have negatively impacted intervention outcomes.

It is recommended that process evaluation protocols are reported and published to encourage transparency.[17–20] The protocol described here builds on emerging process evaluation methodology literature, which can inform the development and implementation of complex interventions.

### Trial status

At the time of submission, the IPCAS trial recruitment is ongoing, with the target sample size of 920 surpassed in March 2019. The IPCAS process evaluation started in March 2018 alongside general practice recruitment, with the fidelity of design and training data collection and analysis commencing in July 2018. Pilot observations and interviews were conducted in February–March 2019, with ongoing iterative analysis.

**Contributors** JM, RM and SS were responsible for obtaining the funding for this study. The design of the process evaluation was led by MRJA, with contributions from JM, RM, EK, VJ and LL. JM, RM, EK, VJ and LL contributed to developing intervention fidelity measures. EK, VJ, LL and JG contributed to piloting intervention fidelity materials, participant recruitment, data collection and analysis. MRJA wrote the first draft of the manuscript. All authors contributed to and approved the final version.

**Funding** This study is funded by the National Institute for Health Research's (NIHR) Programme Grant for Applied Research titled 'Developing primary care services for stroke survivors' (reference PTC-RP-PG-0213–20001). The views expressed are those of the authors and not necessarily those of the NIHR or the Department of Health and Social Care. JM is an NIHR Senior Investigator.

**Competing interests** None declared.

**Patient consent for publication** Not required.

**Provenance and peer review** Not commissioned; externally peer reviewed.

**ORCID iD**
Maria Raisa Jessica (Ryc) Aquino http://orcid.org/0000-0002-3989-1221

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
