## [Reviewer comments · BMJ Open]

ARTICLE DETAILS

TITLE (PROVISIONAL)	Protocol for a multidimensional process evaluation of the Improving Primary Care After Stroke (IPCAS) randomised controlled trial
AUTHORS	Aquino, Maria Raisa Jessica (Ryc); Mullis, Ricky; Kreit, Elizabeth; Johnson, Vicki; Grant, Julie; Lim, Lisa; Sutton, Stephen; Mant, Jonathan

VERSION 1 – REVIEW

REVIEWER	Avril Mansfield University Health Network, Canada
REVIEW RETURNED	07-Feb-2020

GENERAL COMMENTS	Overview/major comments This paper describes a protocol for a process evaluation of a randomized controlled trial, rather than the protocol for the RCT itself. I have no major concerns with the protocol. However, I do question the value of publishing the protocol for the process evaluation at this time, rather than the protocol for the main RCT. While it is not explicitly stated in the submission guidelines that BMJ Open only considers publishing protocols for RCTs (not other study designs), the submission guidelines require that protocols adhere to the SPIRIT checklist, which implies that protocols must be for RCTs. As this is not a protocol for an RCT, the authors have not followed the SPIRIT guidelines/checklist, which makes the paper very difficult to follow. Furthermore, recruiting is already complete for the study (target sample size reached in March 2019), and the expected end date for the study, according to the trial registry, is March 2020. Therefore, the study is almost complete; BMJ Open recommends publishing study protocols in the early stages of study conduct. Minor comments 1. There is no objective in the Abstract.2. The Methods section of the abstract does not describe participants, outcomes etc.3. There is reference to Figure 1 in the paper. Several appendices were attached with the protocol, but it is not clear which of these is Figure 1.
---

REVIEWER	Birgitta Langhammer Faculty of Health Sciences, Oslo Metropolitan University
REVIEW RETURNED	14-Feb-2020

GENERAL COMMENTS	This is a very complex trial and it focuses on a very important issue so it will be interesting to read the results. The trial is
---

	ongoing which hampers the input these comments may have on the protocol. From the protocol however it is difficult to grasp what is actually being done since the package is only shortly described and there is a Reference for it. I cannot find the figure 1 that is also referred to. But a description should also be made here in order to evaluate the Methods section properly. I think the difference between intervention and Control are diffuse and difficult to understand as it is described. the primary end-point are two -subscales of SIS emotion and handicap(!) I thought the description handicap was not in use any longer? And Fidelity to design, training, delivery and receipt and delivery seems more like a feasibility study than an RCT. The involvement of patients / users seems a bit arbitrary, considering the fact that it is improvement of services that persons With stroke have been clamoring. I would have been expecting a closer Collaboration With stroke organisations. However when this is said I think the protocol describes a very complex intervention, the data retrieved will be important in terms of feasibility of a community based service.
--	---

VERSION 1 – AUTHOR RESPONSE

Reviewer 1 comments	Response	Location in manuscript
1. This paper describes a protocol for a process evaluation of a randomized controlled trial, rather than the protocol for the RCT itself. I have no major concerns with the protocol.	Thank you for your feedback on this protocol manuscript.	N/A
2. However, I do question the value of publishing the protocol for the process evaluation at this time, rather than the protocol for the main RCT.	Thank you for this comment. The protocol for the main RCT has been published: Mullis, R., Aquino, M. R. J., Dawson, S. N., Johnson, V., Jowett, S., Kreit, E., & Mant, J. (2019). Improving Primary Care After Stroke (IPCAS) trial: protocol of a randomised controlled trial to evaluate a novel model of care for stroke survivors living in the community. BMJ open, 9(8), e030285.	N/A

	We see the value in publishing this protocol at this time, following the publication of the main RCT protocol, in line with UK Medical Research Council guidance where publication(s) reporting process evaluation findings should refer to ‘a protocol paper or report that clarifies how the component publications relate to the overall evaluation’ (Moore et al., 2015; p. 7).	
--	--	--

3. While it is not explicitly stated in the submission guidelines that BMJ Open only considers publishing protocols for RCTs (not other study designs), the submission guidelines require that protocols adhere to the SPIRIT checklist, which implies that protocols must be for RCTs. As this is not a protocol for an RCT, the authors have not followed the SPIRIT guidelines/checklist, which makes the paper very difficult to follow.	Thank you for raising this point. At this stage there is no available checklist for reporting process evaluation protocols. As such, we used guidance written by Moore et al. (2015) in combination with guidance on intervention fidelity written by Bellg et al. (2004). Moore, G. F., Audrey, S., Barker, M., Bond, L., Bonell, C., Hardeman, W., ... & Baird, J. (2015). Process evaluation of complex interventions: Medical Research Council guidance. BMJ, 350. Bellg, A. J., Borrelli, B., Resnick, B., Hecht, J., Minicucci, D. S., Ory, M., ... & Czajkowski, S. (2004). Enhancing treatment fidelity in health behavior change studies: best practices and recommendations from the NIH Behavior Change Consortium. Health Psychology, 23(5), 443. To facilitate ease of reading, we have added a summary of how the rest of the paper is set out: Each of these dimensions – design; training; delivery; engagement (enactment and receipt) – are discussed in turn, including relevant materials and data collection processes. These are followed by the patient and public involvement and analysis plans.	Page 4, lines 159-162 Table 1_2020_04-revised (page 5, line 168)
---	---	--

4. Furthermore, recruiting is already complete for the study (target sample size reached in March 2019), and the expected end date for the study, according to the trial registry, is March 2020. Therefore, the study is almost complete; BMJ Open recommends publishing study protocols in the early stages of study conduct.	Thank you for raising this concern. The study has been extended to September 2020 and the date on the trial registry has since been updated: https://clinicaltrials.gov/ct2/show/NCT03353519. We appreciate that the study is not in the early stages of conduct, however, this is an ongoing study which still fits the remit of BMJ Open protocol manuscripts. Importantly, we believe it is critical to publish the process evaluation protocol prior to data lock; i.e., before analysis is carried out.	N/A
5. There is no objective in the Abstract.	Thank you - we have clarified the aim/objective in the abstract: The process evaluation aims to: explore how the intervention was delivered in context, and how participants engaged with the intervention.	Page 2, line 62
6. The Methods section of the abstract does not describe participants, outcomes etc.	Thank you for this comment. We have revised the methods to clarify sampling and participants: Methods and analysis A mixed-methods design will be used. 1) Design: Intervention content will be compared to 'usual care'. 2) Training: Intervention training sessions will be audio/video-recorded where feasible 3) Delivery: Healthcare professional self-reports, audio-recordings of intervention delivery and observations of MLAS course (10% of reviews and sessions) will be coded separately; semi-structured interviews will be conducted with a purposive sample of healthcare professionals. 4) Receipt and 5) Enactment: Where available, structured stroke review records will be analysed quantitatively; semi-structured interviews will be conducted with a purposive sample of study participants. Self-reports, observations and audio/video recordings will be coded and scored using specifically developed checklists. Semi-structured interviews will be analysed thematically. Data will be analysed iteratively, independent of primary endpoint analysis.	Page 2, lines 66-72

7. There is reference to Figure 1 in the paper. Several appendices were attached with the protocol, but it is not clear which of these is Figure 1.	Thank you for highlighting this; we have updated the file with a visible figure label.	Attached PDF: Figure 1_2020_04-revised
Reviewer 2 comments	Response	Location in manuscript
1. This is a very complex trial and it focuses on a very important issue so it will be interesting to read the results. The trial is ongoing which hampers the input these comments may have on the protocol.	Thank you for your feedback on this protocol manuscript.	N/A
2. From the protocol however it is difficult to grasp what is actually being done since the package is only shortly described and there is a Reference for it.	Thank you for this point. We have revised the manuscript to include further details on the intervention: The Improving Primary Care After Stroke (IPCAS) cluster randomised controlled trial (RCT) aims to evaluate the clinical and cost effectiveness of a new model of primary care for stroke survivors living in the community compared with standard care, and has been described previously (12). General practices randomised to the intervention arm will deliver a multicomponent package of care comprising: a one-off structured review with a trained primary care professional using a novel needs checklist (13), a self-management course (My Life After Stroke; MLAS) conducted by trained facilitators over a nine-week period, which aims to improve confidence, independence and hope in survivors of stroke through problem-solving and experience and knowledge exchange with course participants; enhanced communication between healthcare professionals, and a direct point of contact in primary care accessible to patients for stroke-specific support throughout the study period. In addition, Figure 1 is now revised, reflecting a visible figure label which depicts the intervention components and its outcomes.	Page 3, lines 109-119 Attached PDF: Figure 1_2020_04-revised

3. I cannot find the figure 1 that is also referred to.	Thank you for highlighting this; we have updated the file with a visible figure label.	Attached PDF: Figure 1_2020_04-revised
4. But a description should also be made here in order to evaluate the Methods section properly.	See response to point 2.	N/A
5. I think the difference between intervention and Control are diffuse and difficult to understand as it is described.	Thank you for this. Further information regarding the control group is now given: General practices randomised to the control arm will provide 'usual care'; the details of which will be documented as part of this process evaluation (see Fidelity of Design).	Page 3, lines 119-120 Attached PDF: Figure 1_2020_04-revised
6. the primary end-point are two -subscales of SIS emotion and handicap(!) I thought the description handicap was not in use any longer?	Thank you for this query. The Stroke Impact Scale identifies 'handicap' as one of its domains. We appreciate that this term might be outdated, however, we need to use the original domain terms given that this is a standardised questionnaire: Jenkinson, C., Fitzpatrick, R., Crocker, H., & Peters, M. (2013). The Stroke Impact Scale: validation in a UK setting and development of a SIS short form and SIS index. Stroke, 44(9), 2532-2535.	N/A
7. And Fidelity to design, training, delivery and receipt and delivery seems more like a feasibility study than an RCT.	Thank you for this comment. Various dimensions or aspects of intervention fidelity are used in this mixed-methods process evaluation to assess the extent to which the intervention (i.e. IPCAS, which is being tested in a cluster randomised controlled trial; Mullis et al., 2019) is delivered and applied as designed.	N/A

8. The involvement of patients / users seems a bit arbitrary, considering the fact that it is improvement of services that persons With stroke have been clamoring. I would have been expecting a closer Collaboration With stroke organisations.	Thank you for raising this point. The intervention (IPCAS) was developed in close collaboration with stroke stakeholders (Mullis et al., 2019). The materials for this process evaluation, also involved patient and public involvement (PPI) input: We collaborated with our long-standing PPI member regarding the design of all patient-facing materials, and piloted the interview schedule with them. This ensured that all communications utilised accessible and jargon-free language, and data collection procedures were suitable for survivors of stroke. We acknowledge that our PPI input for the process evaluation is less involved than the intervention development phase, however, it was focussed and targeted to ensure meaningful and collaborative involvement. This is now reflected in the manuscript: A PPI member provided feedback on the process evaluation protocol to ensure procedures (e.g. recruitment, data collection) in place are appropriate and relevant. For example, PPI consultation resulted in changes to the format and length of the questionnaires for MLAS, ensuring that the questionnaire is aphasia friendly. Together, we decided to brief facilitators about providing stroke survivors support in completing questionnaires where needed. Our PPI member was also involved in pilot-testing the interview schedule with a member of the research team, which ensured that the interview schedule was suitable and flexible (e.g. addition of closed question prompts for people with aphasia).	Page 9, lines 303-310
9. However when this is said I think the protocol describes a very complex intervention, the data retrieved will be important in terms of feasibility of a community based service.	Thank you for your feedback.	N/A

VERSION 2 – REVIEW

REVIEWER	Birgitta Langhammer Oslo Metropolitan University Norway
REVIEW RETURNED	27-Apr-2020
GENERAL COMMENTS	Comments IPCAS protocol bmjopen-2020-036879.R1:

	This is an ongoing study where they have recruited almost 1000 patients for the RCT. It is not easy to follow the logic in this very complex protocol, but Table 1 was helpful. My primary concern is that the protocol of the evaluation process seems to be created AFTER the primary study was launched which is a bit puzzling? My secondary concern is the complicated method, and the fact that the study is already “on its way” which may influence how people report. Furthermore, the mixed method approach and the huge sample and the few interviewees are not proportional. One may question the recruitment of the interviewees which may give rise to confounders. When this is said, research on interventions and follow up in primary care are very few and this research is needed. The RCT evaluating the model IPCAS will be interesting to follow and of course the evaluation of the process will strengthen the robustness of the model.
--	---

VERSION 2 – AUTHOR RESPONSE

Reviewer 2 comments	Response	Location in manuscript
This is an ongoing study where they have recruited almost 1000 patients for the RCT. It is not easy to follow the logic in this very complex protocol, but Table 1 was helpful.	Thank you for your feedback on our manuscript. To guide the reader, we added further signposting throughout the manuscript: Fidelity of design: ... maintaining a written record of these in the Fidelity of Delivery assessments (detailed in the following subsections). Fidelity of delivery: ... and documented through conducting interviews. A purposively selected sample (approximate n=15; 10 intervention and 5 control) of healthcare professionals will be invited to participate in semi-structured interviews... (see Analysis plan). Fidelity of receipt and enactment (engagement): ...(see Analysis plan).	Page 6, lines 191-192 Page 7 lines 238-247 Page 8 line 294

My primary concern is that the protocol of the evaluation process seems to be created AFTER the primary study was launched which is a bit puzzling?	Thank you for raising this point. The process evaluation protocol was developed alongside the intervention (see Mullis et al., 2019) and is being carried out in parallel to the RCT, as clarified in the manuscript: As such, the present evaluation was developed alongside IPCAS and will be conducted in parallel with it. References: Mullis, R., Aquino, M. R. J., Dawson, S. N., Johnson, V., Jowett, S., Kreit, E., & Mant, J. (2019). Improving Primary Care After Stroke (IPCAS) trial: protocol of a randomised controlled trial to evaluate a novel model of care for stroke survivors living in the community. BMJ open, 9(8), e030285.	Page 4, lines 133-134
My secondary concern is the complicated method, and the fact that the study is already “on its way” which may influence how people report.	Thank you for raising this concern. We developed the process evaluation protocol in response to the need to adequately assess the processes within IPCAS, which is a complex model of care involving patient-level and practice-level components (see Mullis et al., 2019). We utilised a comprehensive approach to protocol development, specifically the Behaviour Change Consortium framework, which provides researchers detailed guidance on assessing various domains of treatment fidelity (see page 4, lines 139-150). We appreciate that the evaluation being “on its way” is a concern. This manuscript remains adherent to BMJ Open’s protocol paper requirements/remit, which includes ongoing pieces of work. Publishing the protocol prior to study completion would be valuable in ensuring that “deviations from the protocol that occur during the conduct of the study” (https://bmjopen.bmj.com/pages/authors/#protocol) are easily identified in subsequent publications. References: Mullis, R., Aquino, M. R. J., Dawson, S. N., Johnson, V., Jowett, S., Kreit, E., & Mant, J. (2019). Improving Primary Care After Stroke (IPCAS) trial: protocol of a randomised controlled trial to evaluate a novel model	N/A

	of care for stroke survivors living in the community. BMJ open, 9(8), e030285. Bellg, A. J., Borrelli, B., Resnick, B., Hecht, J., Minicucci, D. S., Ory, M., ... & Czajkowski, S. (2004). Enhancing treatment fidelity in health behavior change studies: best practices and recommendations from the NIH Behavior Change Consortium. Health Psychology, 23(5), 443.	
Furthermore, the mixed method approach and the huge sample and the few interviewees are not proportional. One may question the recruitment of the interviewees which may give rise to confounders.	Thank you for this comment. In our protocol we provided approximated sample sizes for the qualitative interviews, which will be guided by thematic saturation within our analytical frame. In particular, data collection will cease when relevant information (i.e., in relation to the questions: how was the intervention delivered, and how was the intervention engaged with?) is collected from the participant sample. Therefore, the sample size given is not fixed, and will be finalised within the data collection process. We have reflected this in the manuscript: Fidelity of delivery: ... Recruitment and data collection will cease when adequate data addressing our evaluation aim have been acquired (25,26) (see Analysis plan). Fidelity of receipt and enactment (engagement): ...Recruitment and data collection will cease when adequate data addressing our evaluation aim have been acquired (25,26) (see Analysis plan). In addition, we proposed to use a random purposive sampling strategy for interviews with stroke survivors, which will allow us to capture a variety of views and experiences of providing and engaging with the IPCAS study, thereby addressing potential confounders such as sampling and selection bias. This is reported in page 8, lines 287-289.	Page 7, lines 246-247 Page 8, lines 293-294

When this is said, research on interventions and follow up in primary care are very few and this research is needed. The RCT evaluating the model IPCAS will be interesting to follow and of course the evaluation of the process will strengthen the robustness of the model.	Thank you for your encouraging feedback on our work.	
---	---	--

VERSION 3 – REVIEW

REVIEWER	Birgitta Langhammer Oslo Metropolitan University, Norway
REVIEW RETURNED	06-May-2020
GENERAL COMMENTS	no further comments- goodluck with the trial!